# Computational Analysis of Genetic Code Variations Optimized for the Robustness against Point Mutations with Wobble-like Effects

**DOI:** 10.3390/life11121338

**Published:** 2021-12-03

**Authors:** Elena Fimmel, Markus Gumbel, Martin Starman, Lutz Strüngmann

**Affiliations:** Center for Algorithmic and Mathematical Methods in Medicine, Biology, and Biotechnology, Mannheim University of Applied Sciences, 68163 Mannheim, Germany; e.fimmel@hs-mannheim.de (E.F.); m.gumbel@hs-mannheim.de (M.G.); l.struengmann@hs-mannheim.de (L.S.)

**Keywords:** genetic code, point mutations, wobble effect, evolutionary algorithm

## Abstract

It is believed that the codon–amino acid assignments of the standard genetic code (SGC) help to minimize the negative effects caused by point mutations. All possible point mutations of the genetic code can be represented as a weighted graph with weights that correspond to the probabilities of these mutations. The robustness of a code against point mutations can be described then by means of the so-called conductance measure. This paper quantifies the wobble effect, which was investigated previously by applying the weighted graph approach, and seeks optimal weights using an evolutionary optimization algorithm to maximize the code’s robustness. One result of our study is that the robustness of the genetic code is least influenced by mutations in the third position—like with the wobble effect. Moreover, the results clearly demonstrate that point mutations in the first, and even more importantly, in the second base of a codon have a very large influence on the robustness of the genetic code. These results were compared to single nucleotide variants (SNV) in coding sequences which support our findings. Additionally, it was analyzed which structure of a genetic code evolves from random code tables when the robustness is maximized. Our calculations show that the resulting code tables are very close to the standard genetic code. In conclusion, the results illustrate that the robustness against point mutations seems to be an important factor in the evolution of the standard genetic code.

## 1. Introduction

### 1.1. Motivation

The origin and the structure of the standard genetic code (SGC), i.e., the codon–amino acid assignment is still a subject under research [1]. There are at least three major theories (overview in [2]): (1) The stereochemical theory states that codon assignments for specific amino acids are determined by physicochemical affinities between amino acids and cognate codons or anticodons [3]. This hypothesis is not very well supported by experimental data, though. (2) The coevolution theory explains the structure of the standard code by pathways of amino acid biosynthesis which were added step by step [4,5]. (3) A third theory (adaption theory) claims that the SGC has evolved under selective pressure to minimize translation errors [6]. In particular, the code is robust against point mutations under this assumption.

All three theories agree to some extent that evolution has included more amino acids to be encoded [7,8]. Understanding the principles of the genetic code enables the development of a modified SGC with non-canonical amino acids [9,10]. This is of importance in biotechnology as, for instance, it facilitates the development of novel drugs. Recently, Błażej and colleagues analyzed the extension of the SGC that was inspired by the robustness of the code against point mutations [11].

A graph-based approach was presented in our previous work [12] which quantifies for a (generalized) genetic code its robustness against point mutations. A weighted graph represents possible point mutations for each codon (or tuple in general). We demonstrated that such a genetic code shows better robustness and the weights are related to the wobble effect. This paper continues our work and searches for weighted graphs that maximize the robustness of the standard genetic code.

### 1.2. Conductance of the Genetic Code

This section briefly recalls the definitions from [12] which are based on [13]. Let Σ be a finite alphabet of even cardinality |Σ|=2n for some n∈N. As a special alphabet B = {A, C, G, T(U)} is used for the standard genetic alphabet.

**Definition** **1.**
*Let ℓ∈N and $P=\{p_{i}^{\left\{N,N^{\prime }\right\}}|i=1,\ldots ,\ell ,N\neq N^{\prime }\in \Sigma \}$ where $p_{i}^{\left\{N,N^{\prime }\right\}}$ are non-negative weights. We define a weighted graph $G\left(V,E\right)=G_{l}^{P}\left(V,E,w\right)$ as follows:*
(1)
*$V=\Sigma^{\ell }$ is the set of vertices (nodes) representing all possible ℓ-letter words over *Σ*;*
(2)
*E is the set of edges where $(c,c^{\prime })\in E$ if and only if $c,c^{\prime }\in V$ and c differs from $c^{\prime }$ in exactly one position;*
(3)
*The function w:E→P assigns to every edge $(c,c^{\prime })\in E$ a weight $p_{i}^{\left\{N,N^{\prime }\right\}}$ by $w\left(\left(c,c^{\prime }\right)\right)=p_{i}^{\left\{N,N^{\prime }\right\}}$ if and only if c differs from $c^{\prime }$ in position i∈{1,…,ℓ} and $c_{i}=N,c_{i}^{\prime }=N^{\prime }$.*

*If for all i∈{1,⋯,l} the weights $p_{i}^{\left\{N,N^{\prime }\right\}}$ are independent of the choice of $N,N^{\prime }$ we will simply denote the weights $p_{i}^{\left\{N,N^{\prime }\right\}}$ by $p_{i}$.*


According to Definition 1 the graph *G* is weighted, undirected, and regular. Note that the weights are symmetric, i.e., $p_{i}^{\left(N,N^{\prime }\right)}$ = $p_{i}^{\left(N^{\prime },N\right)}$. Given Σ=B = {A, C, G, T(U)} the graph *G* has a biological interpretation: The set of edges *E* represents all possible single point mutations, which can occur between codons in protein-coding sequences. Such point mutations appear quite often and might lead to fatal changes in translated proteins (see [14]). The weights $p_{i}$ can be considered, if standardized, as the probabilities with which a point mutation occurs at position *i*. This definition also takes into account that these mutation probabilities may depend not only on the position in the codon, but also on the base pairs. For example, it is likely that the mutation U → G in the third position of a codon occurs more frequently than the mutation U → A [15]. Figure 1 depicts an example of a graph that satisfies Definition 1.

Given this definition, we might ask how the set of all available codons (*ℓ*-tuples in general) could be partitioned into disjoint subsets where each subset represents an amino acid to be encoded and where the influence of single point mutations is minimal. This can be achieved by solving a graph clustering problem.

Let $C_{k}$ be a partition of nodes of the graph *G* into a fixed number $1<k\leq {\left(2n\right)}^{\ell }$ of disjoint non-empty subsets $C_{k}$:$$C_{k}=\left\{S_{1},S_{2},\ldots ,S_{k}:\;S_{i}\cap S_{j}=\emptyset ,\;S_{1}\cup S_{2}\cup \ldots \cup S_{k}=V\right\}.$$

Each subset $S_{i}$ in this partition is supposed to contain codons that encode the same amino acid. According to [13] the “quality” of such a partition can be measured by means of a so-called conductance. We first define the conductance for a single subset *S* of *V* [16] and adapt then the definition from [13] to weighted graphs:

**Definition** **2.**
*For a given weighted graph G=G(V,E,w) let S be a subset of $V=\Sigma^{l}$. We define the set-conductance of S as:*

$$\phi \left(S\right)=\frac{w(E\left(S,\bar{S}\right))}{\sum_{c\in S,(c,c^{\prime })\in E}w\left(\left(c,c^{\prime }\right)\right)}$$

*where $w(E\left(S,\bar{S}\right))$ is the sum of the weights of edges of G crossing from S to its complement $\bar{S}$:*

$$E\left(S,\bar{S}\right):=\{\left(c,c^{\prime }\right)\in E:|\left\{c,c^{\prime }\right\}\cap S\left|=1\right\}.$$


*We set $w(E\left(S,\bar{S}\right)):=0$ if $E(S,\bar{S})=\emptyset$. As an extension, the set-robustness ρ(·) is additionally defined as*

ρ(S):=1−ϕ(S).



Figure 2 shows an example for Definition 2. In a biological context, ϕ(S) has a useful meaning. If all codons of *S* encode the same amino acid or the stop signal, then ϕ(S) is the ratio of the number of non-synonymous single nucleotide substitutions to all possible nucleotide substitutions. Analogously, ρ(S)=1−ϕ(S) represents synonymous mutations and thus the robustness of a cluster of codons against point mutations. An ideal code, in this context, would have a conductance ϕ(S)=0 or, equivalently, a robustness ρ(S)=1. Note that some greater weight $p_{i}^{\left\{N,N^{\prime }\right\}}$ of one edge in $E(S,\bar{S})$ (see Definition 1) means more robustness against point mutations in the sense that $p_{i}^{\left\{N,N^{\prime }\right\}}$ increases $w(E\left(S,\bar{S}\right))$ in the numerator of ϕ(S). Hence ϕ(S) will increase as the sum off all weights in the denominator contains $w\left(E\left(S,\bar{S}\right)\right)+w\left(E\left(S,S\right)\right)$ whereas the weights in w(E(S,S)) remain the same.

The conductance for a partition was proposed in [17], adopted in [13,18] and is defined as the conductance of the “weakest link in the chain”:

**Definition** **3.**
*For a given weighted graph G=G(V,E,w) the conductance of a partition $C_{k}$ of $V=\Sigma^{l}$ is defined as*

$$\Phi \left(C_{k}\right)=\max_{S\in C_{k}}\phi \left(S\right).$$



Φ gives a characterization of the quality of a partition $C_{k}$ as the set conductance of the worst *l*-letter group in this partition. However, there might be a problem: If an amino acid-like Methionine is encoded by exactly one codon (like in the standard genetic code) we always have $\Phi (C_{k})=1$ no matter how good the other set-conductances are. This can be mitigated by the definition of average conductance.

**Definition** **4.**
*The average conductance of a partition $C_{k}$ is defined as*

$$\bar{\Phi }\left(C_{k}\right)=\frac{1}{k}\sum_{S\in C_{k}}\phi \left(S\right).$$



In addition to the average conductance, there is also the average robustness.

**Definition** **5.**
*The average robustness of a partition $C_{k}$ is defined as*

$$\bar{P}\left(C_{k}\right)=1-\bar{\Phi }\left(C_{k}\right)$$


*or equally*

$$\bar{P}\left(C_{k}\right)=\frac{1}{k}\sum_{S\in C_{k}}\rho \left(S\right).$$



The average conductance for the SGC with all weights set to 1 is $\bar{\Phi }\left(C_{\mathrm{SGC}}\right)$≃0.81 [19]. It can be decreased to $\bar{\Phi }\left(C_{\mathrm{SGC}};P_{M}\right)$≃0.54 when the weights are set according to Table 1 which was proven in [12] (Table 1). However, it is an open question whether these weights are already optimal.

In this paper, we search for optimal weights that minimize the average conductance $\bar{\Phi }$. We also look for a genetic code table that minimizes the average conductance while the weights in Table 1 are fixed.

## 2. Materials and Methods

This section introduces the methods for the optimization of the conductance according to Definition 1. We show three possible ways to approach this topic.

(1)The first problem is the optimal adjustment of the weights to minimize the conductance of the SGC (this task is abbreviated as and later referred to *EA weights*). In Section 2.1 we describe the optimization algorithm used to achieve these objectives.(2)The second attempt is to find a genetic code table with 21 classes such that the average conductance is optimal with respect to the weighting of the graph according to (a) Table 1 and (b) the optimal weights found in (1) (task *EA code table*).(3)Finally, we discuss if both the weights and the structure of genetic code, are optimized.

Subsequently, in Section 2.2 and Section 2.3 the adjustments of the EA for the three methods are outlined. Concluding, Section 3.1, Section 3.2, Section 3.3 and Section 4 discuss the results obtained.

### 2.1. Evolutionary Algorithm

We choose an *evolutionary algorithm* (EA; also *genetic algorithm*) to solve the optimization problem [20,21]. EAs are algorithms suitable for discrete optimization problems inspired by Darwin’s theory of evolution. In the simplified model of Darwin’s theory used in EAs, a population evolves through mating and mutation to adapt to the environment. This process consists of crossing over the individuals in the population followed by a certain number of mutations within the population and a final selection of the fittest member (see Figure 3).

The EA only provides a framework for optimization. The number of individuals that are mated or mutated is parameterized and can be set according to the application. Additionally, a factor can be set to control the survival rate after each iteration. The algorithm also demands the definitions of the crossover and mutation functions and a function to obtain a numerical fitness value. The Δ parameter controls the termination condition. It is defined in such a way that if the change of the fitness of the best individual in the population in the last optimization step is smaller than Δ then the condition is fulfilled and the EA is terminated. Our implementation is based on the R package GA (short for: *genetic algorithm*) [22].

### 2.2. Optimal Weights

In order to optimize the weights (task *EA weights*), they must first be put into a suitable form. Table 1 illustrates the weights as an example. A weight $p_{i}^{\left\{N,N^{\prime }\right\}}$ at an edge representing a mutation is specified by the position of the mutation and the type of mutation. The position *i* of the mutation indicates which matrix must be used to select the weight. This can be identified by the position index in the upper left corner of the matrix. The type of the mutation $\{N,N^{\prime }\}\in B$ gives the row and the column and therefore the actual weight. While the regular base *N* is specified by the row index, the mutated base $N^{\prime }$ is specified by the column index. From Definition 1 it follows that elements on the diagonal are undefined and we set them to 0. Since the graph is undirected it follows that the values in the matrix are symmetric.

The matrix row-column indexes for every base position according to Table 1 are transformed into a linear list in order to implement the crossover function. The three matrices and the list share the same data and can be used alternately. The used mapping is outlined in Table 2.

For each pair of parents to be mated, a crossover point between one and 18 is randomly selected. The offspring is generated by exchanging the weights of the parents with each other until the crossover point is reached. This procedure relies on the linear index system from Table 2.

The algorithm mutates a randomly chosen position in one of the three matrices. The mutation is represented by the multiplication of a randomly chosen weight and its symmetric element. The factor is a uniform random number between $\frac{1}{1.2}$ and 1.2 which reflects a mutation rate of ±20%.

This fitness for the weight matrices *P* represents the robustness of the genetic code (see Definition 5) with respect to the weight matrix *P*:$$f\left(P\right)=\bar{P}\left(C_{SGC};P\right)$$

The optimization is started with a population of 150 random weight tables. The cross-over rate is 60% and the mutation rate 40%. The optimization is terminated if the increase of the fitness is less than a given threshold of $\Delta =10^{-6}$. According to Definition 1 the weights have to be positive real numbers. It should be noted that some weights could go to infinity and others to zero without further restrictions to the EA, though this did not happen in our optimizations.

Finally, after an optimization process the weights are normalized as the assignment of weights is not unique which is an intrinsic property of the conductance (see Definition 2). The weights can be scaled by any factor. As the 3 matrices contain 36 non-zero values (where actually only 18 can be different) we impose the constraint that the sum of all weights is 36. This is equivalent to a graph where all edges have a weight of 1. We show that this constraint is valid. Let *P* be the weights and *S* any partition.
$$\begin{array}{cc} s & =\sum_{p_{i}^{\left\{N,N^{\prime }\right\}}\in P}p_{i}^{\left\{N,N^{\prime }\right\}}\\ P^{\prime } & =\{p_{i}^{\left\{N,N^{\prime }\right\}}\cdot \frac{36}{s}:p_{i}^{\left\{N,N^{\prime }\right\}}\in P\} \end{array}$$

Then the conductance ϕ(S) is the same with *P* or $P^{\prime }$ which is clear if we look at a simplified equation of ϕ(S). Let x,y∈R be the numerator and denominator of $\phi \left(S\right)=\frac{x}{y}$ with respect of the weight matrices *P*. Then one can simplify the equation ϕ(S) with respect to the new weight matrices $P^{\prime }$ as:$$\phi \left(S\right)=\frac{x\cdot \frac{36}{s}}{y\cdot \frac{36}{s}}.$$

Yet, now the following holds:(1)$$\sum_{p_{i}^{\left\{N,N^{\prime }\right\}}\in P^{\prime }}p_{i}^{\left\{N,N^{\prime }\right\}}=36.$$

### 2.3. Optimal Genetic Code Table

In a reverse approach, we want to optimize the partitioning using given weight matrices (task *EA code table*). More precisely, this method searches for a code table that maximizes the robustness while the weight matrices related to the SGC are fixed. The only constraint we impose is that we decompose the 64 codons of this table always into 21 classes. Note that the number of classes must be fixed and cannot be left to the algorithm as a further variable to be optimized. This constraint results from the fact that a code with a degeneracy of one would always be the optimum and thus the result would always be the same and without significance.

As introduced above we use an EA for this task. Yet, the crossover and mutation operators must be adapted to the additional condition that a partition requires 21 classes. Hence, we have decided that the crossover function, as well as the mutation function, will ensure that the code tables will consist of 21 classes. To ensure that the crossover operator meets the requirements, only one offspring per mating is produced instead of two. This offspring is the result of a simple random mating of the parents. The randomness is only affected by the assurance that the offspring has 21 classes. The implementation of the crossover operator is introduced in the pseudo-code listed in Algorithm 1.
**Algorithm 1** Crossover operator for partitions **function**
CrossoverPartitions($C_{1}$, $C_{2}$)         ▹$C_{1}$ and $C_{2}$ are partitions   $C_{new}\leftarrow copy\left(C_{1}\right)$                 ▹ copy first partion   **for** idx∈1…64 **do**     Let *r* be a uniform random number between 0 and 1     **if** r≥0.5 **then**       $C_{new}\left[idx\right]\leftarrow C_{2}\left[idx\right]$       ▹ assign class from second partition       **if** $C_{new}$ has not 21 classes **then**         $C_{new}\left[idx\right]\leftarrow C_{1}\left[idx\right]$         ▹ undo assignment       **end if**     **end if**   **end for**   **return** $C_{new}$ **end function**

A mutation replaces for a randomly selected codon its associated class with the class of another random codon. To ensure that the code table still has 21 classes, care is taken when selecting the random codon: its class must not occur only once because then the replacement would remove this class permanently.

The fitness function calculates the robustness of the optimized table *C* under the influence of the given weight matrices *P*:$$f\left(C\right)=\bar{P}\left(C;P\right)$$

The optimizer is initialized with 100 individuals (random code tables). The cross-over rate is 60% and the mutation rate is 40%. Like before, the process is terminated if the increase of the fitness function value drops below a threshold of $\Delta =10^{-6}$.

### 2.4. Single Nucleotide Variants

The weights of the graph indicate how severe a point mutation would be. We would like to compare the weights to the number of single nucleotide variants (SNV) that can be found in biological coding sequences (CDS). Mutations in non-coding DNA regions cannot be considered as the translation of codons into amino acids is optimized in our analysis. Nevertheless, it can be expected that the distribution of point mutations in non-coding sequences will be different and that there are patterns in non-coding DNA [23]. The coding sequences were taken from mouse (*Mus musculus*) chromosome 1 and 2 and downloaded from Ensemble’s biomart (http://www.ensembl.org/biomart/martview; accessed on 3 November 2021) [24].

The data base *Mouse Short Variants (SNPs and indels excluding flagged variants) (GRCm39)* was chosen. In particular, the biomart attributes
Variant alleles (e.g., C/U which represents a mutation C → U)Variant start in CDS (bp)Variant end in CDS (bp)
enable us to determine the frequency of mutations per base position within a codon. Rows where (1) the variant start or end is *not a number* (nan) or (2) rows with no point mutations were removed.

The base position i∈{1,2,3} for a codon can be calculated from the attribute *Variant start in CDS (bp)*, denoted as s∈N,s≥1:i=((s−1)mod3)+1

Given the frequencies of mutations per position one can easily create transition matrices for each position (see Table 4). Typically, the transitions represent probabilities or relative frequencies. Our transition matrices are scaled such that the sum of all values equals 36 according to Equation (Equation 1).

## 3. Results

This section elaborates on the results. We begin with the optimal weights (task *EA weights*, Table 3 and Table 4) that were found by the genetic algorithm and then present optimal genetic code tables (*EA code table*).

### 3.1. Optimal Weights

Table 3a shows the weights of Table 1 as used in [12] where the matrices were normalized to have a sum of 36. This will be done for all matrices in the following sections. The conductance for these weights is about $\bar{\Phi }\left(C_{\mathrm{SGC}};P_{M}\right)=0.54$ (rounded to 2 digits) which is better than the conductance of the unweighted graph $\bar{\Phi }\left(C_{\mathrm{SGC}}\right)=0.81$ [19]. The matrices in Table 3b list the optimal weights of the SGC found by the optimization. The average conductance of the SGC with this weight distribution table is $\bar{\Phi }\left(C_{\mathrm{SGC}};P_{\mathrm{opt}}\right)=0.12$. The conductance (or the robustness) could be considerably improved.

It is remarkable that the value $p_{3}^{\left\{U,C\right\}}$ (or mutation U ↔ G in the third position) in Table 3b is the highest and the second-highest is $p_{3}^{\left\{A,G\right\}}$ (or A ↔ G). These values show wonderfully the influence of the wobble effect on the structure of the SGC which is reflected by this conductance measure. These results also suggest that the codon position can be classified according to its vulnerability to any point mutations or mistranslation at the ribosome: The weights in the matrix for position 2 are the smallest (average of 0.0033), followed by position 1 (average of 0.0037) and the highest values can be found at position 3 (average of about 3).

This order is the same when we look at point mutations found in coding sequences which confirms the results of our model. Figure 4 shows the relative frequencies of point mutations in coding sequences according to their position in a codon in mouse chromosome 1 and 2. Clearly, most of these mutations occur at position 3 which supports the findings of the conductance weights in Table 3a,b. Moreover, the ratio of the number of synonymous (silent) mutations to the number of all point mutations is by far greatest at position 3 (about 90%). At position 1 about 10% of the point mutations are synonymous and position 2 has the smallest value of only 1%. We conclude that the wobble effect in position 3—as pointed out by the weights—promoted mutations but those which do not affect the protein sequence. Mutations at position 1 or in particular 2 will more likely lead to mutations that change the amino acid. As a consequence, those mutations are very rare.

Three transition matrices for the point mutations of the coding sequences were calculated for each base position in the next step (see Table 4a). The relative transitions were (again) normalized such that the sum of all transitions equals 36 in order to be compatible with Table 3a,b. Table 4b shows the same transitions with a symmetrical matrix. This formatting has no influence on the calculation of the conductance and is helpful because the conductance weights are also symmetrical. Though the exact numbers differ from the conductance weights, the tendency is clearly the same. The two highest transitions by far are reached at base position 3 for the transitions G ↔ A with a value of 4.8 (or in the directed form G → A with value 6 and its reverse A → G with value 3.5) as well as for C ↔ U with a value of 4.5 (or C → U with 5.7 and its reverse U → C with 3.3). This is qualitatively consistent with the optimal weights ($P_{\mathrm{opt}}$) where these two transitions also have the highest values. These results are also in line with the well-known fact that point mutations are dependent on their environment in a sequence (e.g., CpG islands) and that as a consequence the transitions U ↔ C and A ↔ G mutations occur more frequently [15]. Transversions like G ↔ C or G ↔ U are less frequent. Jiang and coworkers analyzed single point mutations (SNPs) in chimpanzees [25]. They found for exons that the mutation C→U has a relative frequency of 28.3% and U→C 11.3%. Similarly, the mutation G→A has a relative frequency of 27.3% and A→G 11.7%. All other point mutations have a relative frequency less than 4.2%. These results, again, support our findings.

The evolutionary algorithm found an optimal set of weights that led to an average conductance of about $\bar{\Phi }\left(C_{\mathrm{SGC}};P_{\mathrm{opt}}\right)=0.12$. In the following proposition we show that the lower bound of the average conductance is 0.0953.

**Proposition** **1.**
*Let $C_{\mathrm{SGC}}$ be the partitioning of the SGC and 0≤p≤∞ for all p∈P. Then the lower bounds of the average conductance is $\bar{\Phi }\left(C_{\mathrm{SGC}}\right)>0.0953$.*


**Proof.** It can be shown that the average conductance of the SGC cannot be lower than $\bar{\Phi }\left(C_{\mathrm{SGC}}\right)>0.0953$. This is due to three reasons. Firstly, that Methionine and Tryptophan appear as a partition of size one. Secondly, that the Isoleucine partition of size three disjoints AUA and AUG. Finally, the stop signal which separates UGA and AUG. Hence, we can conclude that the conductance of Methionine and Tryptophan (Try) is one and the conductance of Isoleucine (Ile) and the stop signal is greater than zero, i.e., $x=\phi \left(S_{Try}\right)+\phi \left(S_{Ile}\right)>0$. This is because all weights are $p_{i}^{\left\{N,N^{\prime }\right\}}>0$ by definition and it implies that $\bar{\Phi }\left(C_{\mathrm{SGC}}\right)=\left(1+1+x\right)/21>0.0953$. □

This boundary does not seem to be sharp as the results of the optimization process miss the boundary by about 0.025. It suggests that the SGC is not solely optimized according to the conductance measurement. However, the fact that the results are still very good suggests that the influence of the adaption theory on evolution is not negligible.

### 3.2. Optimal Genetic Code Table

This section shows the results of the partitioning optimizations and Table 5 gives an overview. Any code table will always contain 21 classes or labels: 20 for the amino acids and 1 for stop codons.

Before we continue with the optimization results, let us recall the optimal code table for a graph where all weights were set to 1. Such an optimal code table (denoted as $C_{21}$) was developed in [13] and consists of 1 fourfold degenerated group of codons (label 6 in Table 5a) and 20 groups of threefold degenerated codons. Table 5a depicts a minor modification of this code table. The classes labeled as 2, 8, 12, and 18 are the bottom of their blocks of four codons whereas the table in [13] shows these labels at the top of each block. i.e., codon UUG with label 2 would be UUU in the original table. Both tables, the original one and the one shown here, have the same average conductance of $\bar{\Phi }\left(C_{21}\right)=146/189$≃0.77. The modification makes the original code table more compatible to the SGC with respect to potential mappings to amino acids. Nevertheless, the codons of this table can only poorly be mapped to corresponding amino acids. Exceptions are the block AU* with the start codon AUG, which are Methionine and AUU, AUC, and AUA, which can be assigned to Isoleucine, and UUG which encodes for Tryptophan.

Let us move on to the optimization results. Table 5b lists the optimal table where the weights were set according to Table 1 and Table 5c the optimal table which is based on optimal weights according to Table 3b. Interestingly, both code tables are divided into eleven classes of degeneration size four and ten classes of size two. Each table has three classes that cannot be assigned to amino acids in the SGC. These are in Table 5b the classes 4, 14, and 17 and in Table 5c 4, 12, and 17. Yet, the amino acids Leucine, Serine, and Arginine are each represented in two classes in the result tables.

Apparently, a graph with weights that reflect the wobble effect naturally leads to a genetic code table which is quite similar to the SGC. Moreover, the average conductance for the partition $C_{M}$ is $\bar{\Phi }\left(C_{M};P_{M}\right)=0.56$ (rounded to two digits) and the best, i.e., minimal, set-conductance is 0.43. Given the weights $P_{M}$, the conductance could further be improved from 0.77 to 0.56. The average conductance for partition $C_{\mathrm{opt}}$ is $\bar{\Phi }\left(C_{\mathrm{opt}};P_{\mathrm{opt}}\right)=7.7\times 10^{-3}$ and the minimal set-conductance is $1.9\times 10^{-3}$. These values are even smaller. However, this is mainly caused by the optimized and thus better weights. It does not imply that the structure of the code table is better. Indeed, the tables for $C_{M}$ and $C_{\mathrm{opt}}$ are more or less identical as mentioned above.

It should be noted that the partition $C_{M}$ is only one of many possible optimization results. The EA described in the sections above, using the weights $P_{M}$, found also alternative results with the same robustness. These are tables with the same structure, i.e., tables which are permutations of the 16 row blocks which contain either one class of degeneration of size four or two classes of degeneration of size two. Consequently, the table shown is only one of many results of the EA. Yet, other results would have less in common with the SGC. The one presented here was chosen because it is one of the best results in terms of the SGC. The result in $C_{\mathrm{opt}}$, however, is the only optimum. Any permutation would lead to smaller robustness. We would like to recall that this statement is true only if we use the weights $P_{\mathrm{opt}}$ (see Table 3b). In conclusion we can observe that even if the tables for $C_{M}$ and $C_{\mathrm{opt}}$ are more or less identical, yet, the weights $P_{\mathrm{opt}}$ reflect the SGC more accurate since $P_{M}$ is too ambiguous.

### 3.3. Results of Optimizing Both: Weights and Structure

In this section, we show that the simultaneous optimization of weights $P_{\mathrm{opt}}$ and partition $C_{\mathrm{opt}}$ leads to a trivial result. Specifically, it will be shown below that, under some assumptions on the structure of the partition *C*, the (positive) weights can be chosen such that $\bar{\Phi }\left(C_{\mathrm{opt}};P_{\mathrm{opt}}\right)$ tends to zero. Let us first formulate an almost obvious observation:

**Proposition** **2.** 
*Let $G_{l}^{P}\left(V,E,w\right)$ be a given weighted graph as in Definition 1, C be a partition with at least two classes and P a weight matrix with $p_{i}^{\left\{N,N^{\prime }\right\}}>0$ for all weights $p_{i}^{\left\{N,N^{\prime }\right\}}\in P$, $N\neq N^{\prime }$. Then the following must be true:*

$$\bar{\Phi }\left(C;P\right)>0$$



**Proof.** Let us assume that $\bar{\Phi }\left(C;P\right)=0$. In that case ϕ(S)=0 for all S∈C. This means that for each S∈C$E(S,\bar{S})=\emptyset$ holds. This implies |C|=1. This is a contradiction to the assumption |C|≥2. □

The following two observations point the way to the construction of an optimal partition that can have an arbitrarily small average conductance if the weights are chosen appropriately:

**Observation** **1.**
*Let $G_{l}^{P}\left(V,E,w\right)$ be a given weighted graph as in Definition 1 with the weight matrix P, C be a partition of V with at least two classes. Let us further define*

E(p)⊂E

*as a set of all edges in E assigned with the weight p∈P. If each $\left(c,c^{\prime }\right)\in E\left(p\right)$ connects only codons which both belong to the same class $c,c^{\prime }\in S$, thus*

$$E\left(S,\bar{S}\right)\cap E\left(p\right)=\emptyset \;for\;all\;S\in C,$$

*then the weight value p never increases the numerator in ϕ(S). Hence, if such a weight p goes towards infinity ϕ(S) converges toward 0.*


**Observation** **2.**
*Let $G_{l}^{P}\left(V,E,w\right)$, P and C be defined as above. We also adopt the definition of $E(p_{i}^{\left\{N,N^{\prime }\right\}})$ as a subset of E where i∈{1,2,3} and $N,N^{\prime }\in \Sigma$. If each cluster $S_{j}\in C$ contains at least one edge $e\in E(p_{i}^{\left\{N,N^{\prime }\right\}})$*

$$E\left(S_{j},S_{j}\right)\cap E\left(p_{i}^{\left\{N,N^{\prime }\right\}}\right)\neq \emptyset$$

*and (see Observation 1)*

$$E\left(S_{j},\bar{S_{j}}\right)\cap E\left(p_{i}^{\left\{N,N^{\prime }\right\}}\right)=\emptyset$$

*then the average conductance $\bar{\Phi }\left(C;P\right)$ can be made arbitrarily small under the condition that all other weights in P except $p_{i}^{\left\{N,N^{\prime }\right\}}$ are chosen small enough.*


In conclusion, we can say that $\bar{\Phi }\left(C_{\mathrm{opt}};P_{\mathrm{opt}}\right)$ must tend to zero if $C_{\mathrm{opt}}$ and $P_{\mathrm{opt}}$ are simultaneously optimized. Hence, the results would be meaningless. Next we present an example to illustrate this fact.

**Example** **2.**
*Let us now show that it is possible to choose the appropriate edge sets that satisfy the conditions of observations 1 and 2 for code tables in Table 5b,c, and the Vertebrate Mitochondrial Code.*

*In all three cases one can observe that if $E\left(p\right)=E\left(p_{3}^{\left\{C,U\right\}}\right)\cup E\left(p_{3}^{\left\{G,A\right\}}\right)$ then $\left(c,c^{\prime }\right)\in E\left(p\right)$ only connect codons which both belong to the same class $c,c^{\prime }\in S_{i}$ and each class $S_{i}$ has at least one edge e∈E(p). Thus, if all weights $p_{i}^{\left\{N,N^{\prime }\right\}}\in P$ that are unequal to $p_{3}^{\left\{C,U\right\}}$ or $p_{3}^{\left\{G,A\right\}}$ are small enough and $p_{3}^{\left\{C,U\right\}}$, $p_{3}^{\left\{G,A\right\}}$ are chosen large enough, $\bar{\Phi }\left(C;P\right)$ can be made arbitrarily small.*


## 4. Discussion

In this work, it was shown how the measure of conductance in relation to the codon–amino acid assignment can be used to comprehend the robustness of the genetic code with wobble-like effects. To be more precise, all possible point mutations of the genetic code were represented in the form of a weighted graph so that the weights are understood as probabilities of individual point mutations. Optimal weights could be found by means of computer optimizations and they confirm the importance of the wobble effect in the context of the SGC. Furthermore, when wobbling is allowed the robustness of the code could be improved. Our analysis of the weights and the mutations in coding sequences (Table 3b and Table 4b) underlines the importance of the second position within a codon: although the absolute number of point mutations is smallest there, they more often lead to serious changes in the process of translation compared to point mutations in other positions. The next position is the first and the third base position is most robust against point mutations and thus most important for error minimization. These results coincide with the 2-1-3 model by Massey [26], which was—in a similar way—first expressed by Taylor [27] or Dragovich [28] and extended in [29]. This hypothesis claims that the evolution of the codons in the genetic code started with the middle base (position 2) in a codon [30]. In a next step, the first base (position 1) was added to the evolutionary process. Now the distance of two codons could be greater than one point mutation (e.g., AAN and UUN differ in two bases). This, according to Massey, led to some sort of error minimization. Eventually, the third base was added and again, more error minimization was possible. Intriguingly, the only constraint for including new amino acids in new tuples (codons) according to this model is that new amino acids have to be chemically similar to existing amino acids and their cognate codons. It could be shown by simulations that more than 20% of randomly generated code tables that met these constraints have better error minimization properties than the SGC.

If the optimal weights were fixed and the structure is optimized, code tables emerge that are quite similar to the SGC. This is not possible when the wobble effect is not taken into account. We speculate that the SGC has been optimized during evolution to minimize the negative effects of point mutations, however, only to a certain extent. Most likely, it is not the only driving force, which can be seen, among other things, from the fact that the conductance measure of the SGC cannot be 0.

Assuming that the evolution of the SGC is driven only by robustness against point mutations and disregarding the main idea of the frozen accident theory that it is very difficult to change the SGC, one consequence could be that optimization is still ongoing. This hypothesis is even more compatible with the Vertebrate Mitochondrial Code, since this code, with exception of the groups of sixfold degenerated codons, is already very similar to the optimal code in Table 3b. However, if one includes other hypotheses in the theory of evolution, the SGC performs very well. For instance, in [31] by Seligmann and Pollock the assignment of the stop signals in the SGC is explained. Although their argumentation was primarily not intended to explain the codon assignment to stop signals, it follows that the three codons have to be at this position in the code table, i.e., the block structure has to be contained. Additionally, the authors Wong et al. argue in [32] that the codon assignments to Methionine and Tryptophan indicate that they are late arrivals supplied by biosynthesis. The arguments presented support the hypothesis that the optimization of the robustness of the genetic code with respect to point mutations can be considered as a driving force in the evolution of SGC, especially when wobble-like effects are included.

Let us briefly summarize the results. Firstly, we quantitatively support the known fact that the block structure of the SGC favors the wobble effect. Secondly, we show that the transitions U↔C and G↔A are more likely to be detected by the error correction mechanisms compared to the transversions G↔U and C↔G. Finally, we show that the optimized probabilities of single synonymous point mutations derived from the structure of the genetic code mirror the frequencies of single point mutations found in mouse coding sequences.

## Figures and Tables

**Figure 1 life-11-01338-f001:**
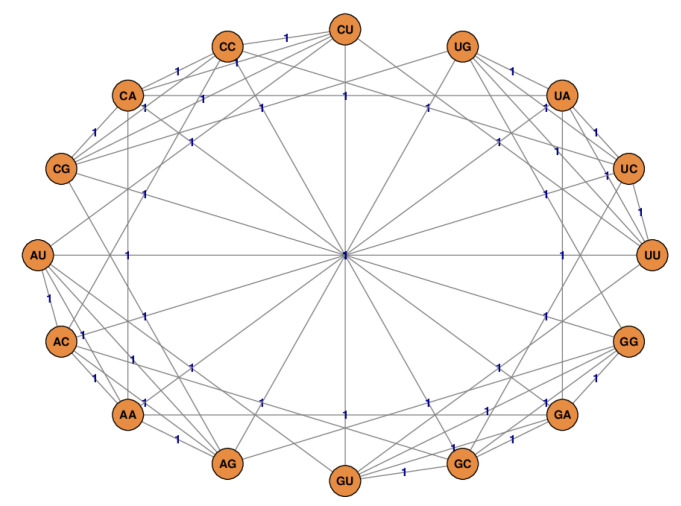
The graph *G* for Σ=B = {A, C, G, U}, ℓ=2 and all weights $p_{i}^{\left\{N,N^{\prime }\right\}}=1$. For simplicity, tuples of length 2 are used instead of codons (ℓ=3)—which leads to only 16 instead of 64 nodes.

**Figure 2 life-11-01338-f002:**
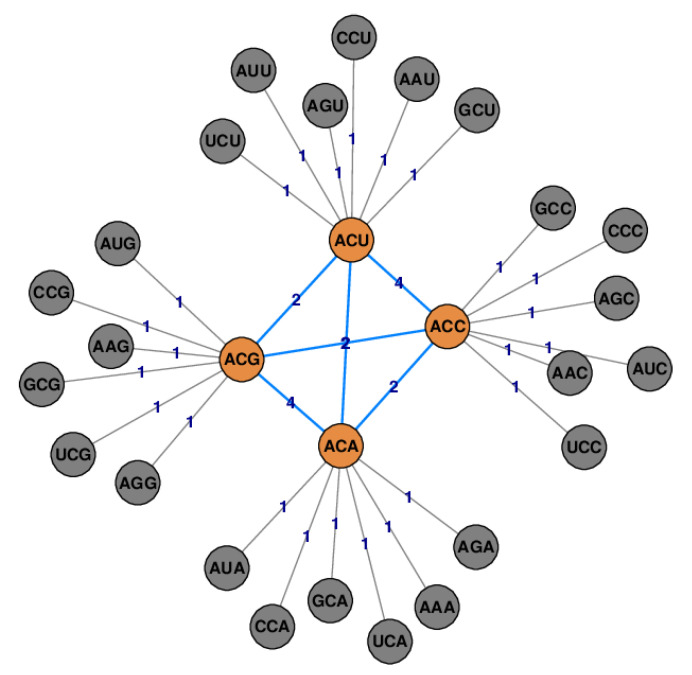
The graph of S = {ACU, ACC, ACA, ACG} inside the graph *G* for Σ=B. The set-conductance is ϕ(S)=24/56≃0.43.

**Figure 3 life-11-01338-f003:**
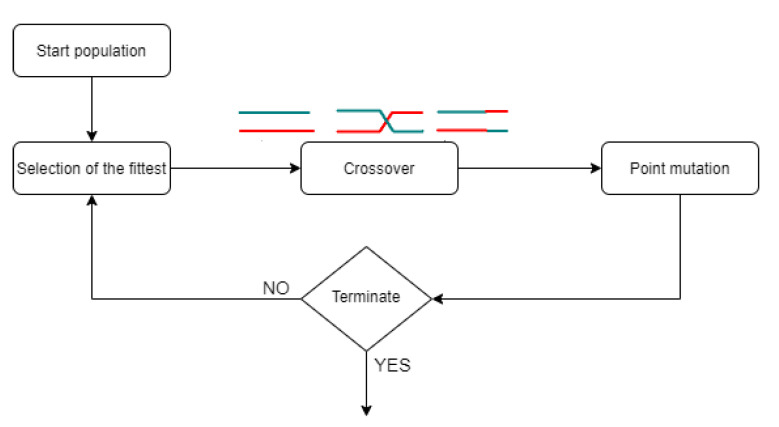
Simplified flow chart of a evolutionary algorithm (EA).

**Figure 4 life-11-01338-f004:**
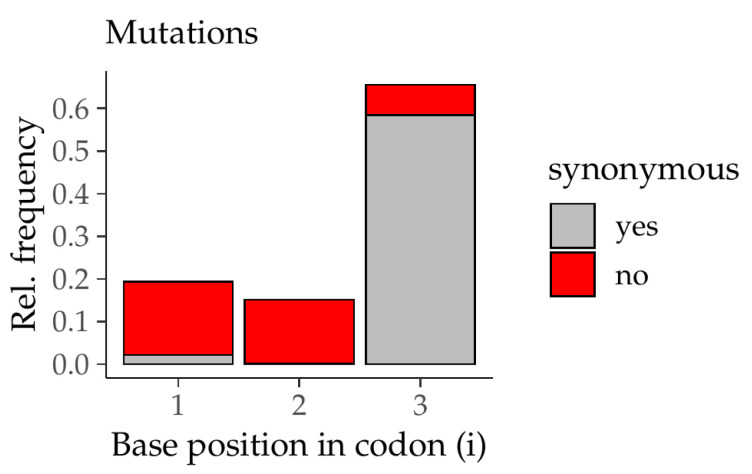
Relative frequencies of point mutations in coding sequences in mouse chromosome 1 and 2 according to their position in the codon. Each bar is divided into non-synonymous (red) and synonymous (or silent; in gray) mutations.

**Table 1 life-11-01338-t001:** Weight distribution $P_{M}$ as introduced in [12]. Each table shows the weights according to base position 1 to 3 within a codon (see number in upper left corner). In accordance with Definition 1 the values are $p_{1}^{\left\{N,N^{\prime }\right\}}=p_{2}^{\left\{N,N^{\prime }\right\}}=1$ for all $N,N^{\prime }\in B$ with $N\neq N^{\prime }$, $p_{3}^{\left\{U,G\right\}}=p_{3}^{\left\{A,C\right\}}=2$ and $p_{3}^{\left\{U,C\right\}}=p_{3}^{\left\{A,G\right\}}=4$.

1	U	C	A	G
U	0	1	1	1
C	1	0	1	1
A	1	1	0	1
G	1	1	1	0
2	U	C	A	G
U	0	1	1	1
C	1	0	1	1
A	1	1	0	1
G	1	1	1	0
3	U	C	A	G
U	0	4	2	2
C	4	0	2	2
A	2	2	0	4
G	2	2	4	0

**Table 2 life-11-01338-t002:** The index transformation from the row-column system to a linear indexing system. (**a**) All non-zero columns are mapped to a linear index ranging from 1 to 18. (**b**) List with values of Table 1.

(a) Indices						
Base position	U ↔ C	U ↔ A	U ↔ G	C ↔ A	C ↔ G	A ↔ G
1	1	2	3	4	5	6
2	7	8	9	10	11	12
3	13	14	15	16	17	18
**(b) Values**						
Base position	U ↔ C	U ↔ A	U ↔ G	C ↔ A	C ↔ G	A ↔ G
1	1	1	1	1	1	1
2	1	1	1	1	1	1
3	4	2	2	2	2	4

**Table 3 life-11-01338-t003:** Optimized weights of the conductance graphs. The values are normalized such that the sum equals 36. (**a**) Weights from Table 1 in normalized form. These weights were used in [12]. (**b**) The optimal weights as found by the evolutionary algorithm. Important figures which are discussed in the text are shown in bold.

(**a**) Normalized weights $P_{M}$ of Table 1				
1	U	C	A	G
U	0	0.6	0.6	0.6
C	0.6	0	0.6	0.6
A	0.6	0.6	0	0.6
G	0.6	0.6	0.6	0
2	U	C	A	G
U	0	0.6	0.6	0.6
C	0.6	0	0.6	0.6
A	0.6	0.6	0	0.6
G	0.6	0.6	0.6	0
3	U	C	A	G
U	0	**2.6**	1.3	1.3
C	**2.6**	0	1.3	1.3
A	1.3	1.3	0	**2.6**
G	1.3	1.3	**2.6**	0
(**b**) Optimal weights $P_{\mathrm{opt}}$ found by evolutionary algorithm				
1	U	C	A	G
U	0	0.006	0.002	0.003
C	0.006	0	0.003	0.003
A	0.002	0.003	0	0.005
G	0.003	0.003	0.005	0
2	U	C	A	G
U	0	0.002	0.002	0.003
C	0.002	0	0.003	0.004
A	0.002	0.003	0	0.005
G	0.003	0.004	0.005	0
3	U	C	A	G
U	0	**15.925**	0.018	0.007
C	**15.925**	0	0.021	0.012
A	0.018	0.021	0	**1.977**
G	0.007	0.012	**1.977**	0

**Table 4 life-11-01338-t004:** Point mutation frequencies. The values are normalized such that the sum equals 36. (**a**) Normalized point mutations per base position of SNV measured in mouse chromosome 1 and 2. The transitions were normalized which makes the comparison with the weights matrix simpler. (**b**) The normalized mutation matrices of (**a**) were made symmetrical as the conductance weights are symmetrical, too. The average value of two corresponding cells is calculated. Important figures which are discussed in the text are shown in bold.

(**a**) Point mutations in mouse CDS				
1	U	C	A	G
U	0	0.9	0.2	0.3
C	1.5	0	0.4	0.3
A	0.2	0.3	0	0.9
G	0.4	0.3	1.4	0
2	U	C	A	G
U	0	0.7	0.2	0.2
C	1.1	0	0.2	0.2
A	0.2	0.2	0	0.7
G	0.2	0.2	1.1	0
3	U	C	A	G
U	0	**3.3**	0.5	0.6
C	**5.7**	0	0.8	0.7
A	0.5	0.6	0	**3.5**
G	0.8	0.6	**6**	0
(**b**) Symmetrical point mutations in mouse CDS				
1	U	C	A	G
U	0	1.2	0.2	0.3
C	1.2	0	0.4	0.3
A	0.2	0.4	0	1.1
G	0.3	0.3	1.1	0
2	U	C	A	G
U	0	0.9	0.2	0.2
C	0.9	0	0.2	0.2
A	0.2	0.2	0	0.9
G	0.2	0.2	0.9	0
3	U	C	A	G
U	0	**4.5**	0.5	0.7
C	**4.5**	0	0.7	0.7
A	0.5	0.7	0	**4.8**
G	0.7	0.7	**4.8**	0

**Table 5 life-11-01338-t005:** Optimized code tables with 21 classes. (**a**) shows the optimal genetic code table as published in [13] where all weights are set to 1. (**b**) is the optimization result if one takes the weights from Table 1. (**c**) is the optimization result if one takes the weights $P_{\mathrm{opt}}$ from Table 3b. Amino acids are displayed when they match the SGC.

(**a**) Optimal partition table $C_{21}$ when all weights are set to 1; $\bar{\Phi }\left(C_{21}\right)$≃0.77									
	U		C		A		G		
U	UUU	**1**	UCU	**7**	UAU	**12**	UGU	**17**	U
	UUC		UCC		UAC		UGC		C
	UUA		UCA		UAA		UGA		A
	UUG	**2**	UCG	8	UAG	**13**	UGG	**18** (Trp/W)	G
C	CUU	**3**	CCU	**9**	CAU	**14**	CGU	**19**	U
	CUC		CCC		CAC		CGC		C
	CUA		CCA		CAA		CGA		A
	CUG	**2**	CCG	**8**	CAG	**13**	CGG	**18**	G
A	AUU	**4** (Ile/I)	ACU	**10**	AAU	**15**	AGU	**20**	U
	AUC		ACC		AAC		AGC		C
	AUA		ACA		AAA		AGA		A
	AUG	**2** (Met/M)	ACG	**8**	AAG	**13**	AGG	**18**	G
G	GUU	**5**	GCU	**11**	GAU	**16**	GGU	**21**	U
	GUC		GCC		GAC		GGC		C
	GUA		GCA		GAA		GGA		A
	GUG	**6**	GCG	**6**	GAG	**6**	GGG	**6**	G
(**b**) Optimal partition table $C_{M}$ for weights $P_{M}$ (see Table 3a); $\bar{\Phi }\left(C_{M};P_{M}\right)$≃0.56									
	U		C		A		G		
U	UUU	**1** (Pha/F)	UCU	**6** (Ser/S)	UAU	**10** (Tyr/Y)	UGU	**17**	U
	UUC		UCC		UAC		UGC		C
	UUA	**2** (Leu/L)	UCA		UAA	**11** (Stop)	UGA		A
	UUG		UCG		UAG		UGG		G
C	CUU	**3** (Leu/L)	CCU	**7** (Pro/P)	CAU	**12** (His/H)	CGU	**18** (Arg/R)	U
	CUC		CCC		CAC		CGC		C
	CUA		CCA		CAA	**13** (Gln/Q)	CGA		A
	CUG		CCG		CAG		CGG		G
A	AUU	**4**	ACU	**8** (Thr/T)	AAU	**14**	AGU	**19** (Ser/S)	U
	AUC		ACC		AAC		AGC		C
	AUA		ACA		AAA		AGA	**20** (Arg/R)	A
	AUG		ACG		AAG		AGG		G
G	GUU	**5** (Val/V)	GCU	**9** (Ala/a)	GAU	**15** (Asp/D)	GGU	**21** (Gly/G)	U
	GUC		GCC		GAC		GGC		C
	GUA		GCA		GAA	**16** (Glu/E)	GGA		A
	GUG		GCG		GAG		GGG		G
(**c**) Optimal partition table $C_{\mathrm{opt}}$ for weights $P_{\mathrm{opt}}$ (see Table 3b); $\bar{\Phi }\left(C_{\mathrm{opt}};P_{\mathrm{opt}}\right)$≃$7.7\;\times \;10^{-3}$									
	U		C		A		G		
U	UUU	**1** (Pha/F)	UCU	**6** (Ser/S)	UAU	**10** (Tyr/Y)	UGU	**17**	U
	UUC		UCC		UAC		UGC		C
	UUA	**2** (Leu/L)	UCA		UAA	**11** (Stop)	UGA		A
	UUG		UCG		UAG		UGG		G
C	CUU	**3** (Leu/L)	CCU	**7** (Pro/P)	CAU	**12**	CGU	**18** (Arg/R)	U
	CUC		CCC		CAC		CGC		C
	CUA		CCA		CAA		CGA		A
	CUG		CCG		CAG		CGG		G
A	AUU	**4**	ACU	**8** (Thr/T)	AAU	**13** (Asn/N)	AGU	**19** (Ser/S)	U
	AUC		ACC		AAC		AGC		C
	AUA		ACA		AAA	**14** (Lys/K)	AGA	**20** (Arg/R)	A
	AUG		ACG		AAG		AGG		G
G	GUU	5 (Val/V)	GCU	9 (Ala/a)	GAU	15 (Asp/D)	GGU	21 (Gly/G)	U
	GUC		GCC		GAC		GGC		C
	GUA		GCA		GAA	16 (Glu/E)	GGA		A
	GUG		GCG		GAG		GGG		G

## Data Availability

The used coding sequences were taken from Mouse (M. musculus) chromosome 1 and 2 and downloaded from http://www.ensembl.org/biomart/martview; accessed on 11/03/2021.

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
