# Peer review of "Computational Analysis of Genetic Code Variations Optimized for the Robustness against Point Mutations with Wobble-like Effects"

_life, 2021, doi:10.3390/life11121338_

Round 1
Reviewer 1 Report
I have the following two comments (suggestions) that should be taken into consideration.
- One of the main results of this manuscript is finding that the second position in codon is the most important with respect to point mutations. The authors connect this finding with 2-1-3 model of evolution of the genetic code (ref. 24). The idea that evolution of the genetic code has evolved in three steps of codon evolution: singlet (2nd position) --> doublet (1st-2nd positions) --> triplet (1st-2nd-3rd positions) has independently introduced in the context of p-adic modeling of the genetic code (B. Dragovich, A. Dragovich: p-Adic Modelling of the Genome and the Genetic Code, Computer Journal, October 2007, DOI:10.1093/comjnl/bxm083), Section 3.2 . It is desirable to take this fact into consideration.
- It would be useful for readers if authors add at the end of Discussion a brief summary listing the main results.
Reviewer 2 Report
The manuscript by Fimmel and co-workers shows that given mutational probabilities reflecting those of coding sequences an optimal (in terms of conductance) genetic code is one with bloc structure (not surprising) but different degeneracies for amino acids. For example, degeneracy of 1 or 6 is not found among the evolved (by an evolutionary algorithm) genetic codes (as we only see few examples of evolved codes, this I only assume). The results fit nicely into accumulating line of evidence that the genetic code was optimized for mutational robustness, but only up to a certain period, and some of the late introductions to the code are not optimal in this sense (e.g. Met, Trp).
I would have looked at more alternative codes, but I understand that there would be as many as there are individual simulations. However, in table 5 we can only look at (a) one that comes from a transition table all set to 1. It shows that it that case the block structure is not confined to position 3, but also to position 1 (in case of type 6) or the 2nd one (in case of type 8). Once the wobble effect of the 3rd position is introduced (from Table 1 or from Table 3), the block structure resembles the SGC to a greater degree. Have you found simulation results to the contrary? What was the average block size? Have you found blocks spanning only 1 triplet, 2 triplets or 5 or more triplets? What is the results of the optimization if using transition matrix from Table 4 (the mouse data)?
When analyzing coding sequences and deriving mutational transitions from that, one needs to keep in mind that only the mutations that do not change the peptides much would be retained. For example, in the discussion (P13L362) the scarcity of mutations in the second position comes from the fact that those would change the coded amino acid to a great degree. Also, because you obtained mutation rates from coding region, it is not intriguing (P10L233) to have wobble-effect and fewer mutations in the 2nd and 1st position. It would be intriguing to find something else.
The wobble effect appears in the title and also prominently discussed in the discussion, but I would already mention in the methodology, that coding sequences from the mouse genome were chosen as the basis for the mutation frequencies so that this block structure of the evolved genetic code would remain so. I expect very different optimal ones (but probably still blocky) if the transition matrix would be based on non-coding sequences (similarly to the code table in table 5a).
The U↔C and G↔A mutations, which are called transitions (as opposed to transversions, like G↔U or C↔G) are more frequent as they can be less efficiently captured by the error correcting mechanisms. Most of the point mutations are due to tautomerization, by which, for example, a C would resemble a U, or an A a G. But pyrimidine bases (C,U) rarely resemble purine bases (A, G).
I do not understand section 3.3 and what does it tell us. Please make it clearer to the mathematically less proficient. It reads more as an afterthought.
I’m not sure I fully agree with the discussion’s conclusion. The SGC is optimal to a certain degree. It could be made more optimal by even a few changes. But one needs to accept that the idea behind the frozen accident theory hold now: it is very difficult to change the SGC. At some point in its evolution the SGC become more and more fixed because more and more peptides dependent on it. Some additions were done, but then the fine tuning to optimization could not proceed, the fitness cost of changing the codon assignment become much larger than the advantage of a more optimal code.
I suggest that manuscript to be accepted for publication with minor changes reflecting my suggestions and concerns above, and the minor points below.
Minor comments
P1L3 Abstract: I would omit the “i.e. they minimize translation errors” part. Translation errors are errors during translation, i.e. the very same mRNA is read differently by the translational mechanism. The right structure of the genetic code can help that, but it also helps the minimalization of mutational errors, that are changes in the genome of the organisms. In this latter sense, different RNAs should be read in the same way (degeneracy) or similarly (having similar amino acids at neighboring codons).
P1L33 I would also mention at least one review from Ned Budisa (Acevedo-Rocha, C.G.; Budisa, N. Xenomicrobiology: A roadmap for genetic code engineering. Microbial Biotechnology 2016, 9, 665–676.) to accompany 9.
P3L81-84: Please elaborate a bit, it was hard to follow for me.
P8L187-190: If you need to keep all 21 types in the code, why not just pick one of the types randomly when mutation a position? When picking from the whole code table, the degeneracies already present influences the probability of getting a new codon assigned to that amino acid, i.e. those amino acids that have more codons already could end up with even more.
P8L198 Mouse do not need to start with a capital letter. The Latin name of the animal, Mus musculus, should be written out in full at first mention.
P8L199: I think the URL of biomart should be put inline with the text and not as a footnode.
P10 Figure 4 is missing
P11 Table 5. The amino acids’ abbreviated names sometimes appear at cells, sometimes not, without any consistency or reason that I see. Would it be possible to include them everywhere? Also, I would color the amino acids, and not just the types in numerical order.
I think than in the case of Copt and Popt and other symbols where the subscripted part is a descriptor (like optimal, minimal, maximal, initial, etc.) and not a variable, it should not be italicized. Only variables are italicized.
P14L367: Make sure that the sentence does not read to suggest only a single nucleotide coded for an amino acid. It was always triplets, but – according to the theory mentioned – only one of the nucleotides (the middle one) had any meaning. And then the first got coding potential and last – and only partly – the third.
In the references, paper titles should not be all capitalized. Only journal names and book titles are capitalized as such.
In reference 1, change the initials of the given names of the authors to Á instead of A. Both of them are called Ádám.
Reference 12 should not have a journal Systems Biology as it is on bioRxiv
